# Can Nanowires Coalesce?

**DOI:** 10.3390/nano13202768

**Published:** 2023-10-16

**Authors:** Vladimir G. Dubrovskii

**Affiliations:** Faculty of Physics, St. Petersburg State University, Universitetskaya Emb. 13B, 199034 St. Petersburg, Russia; dubrovskii@mail.ioffe.ru

**Keywords:** nanowires, coalescence, partial merging, surface coverage

## Abstract

Coalescence of nanowires and other three-dimensional structures into continuous film is desirable for growing low-dislocation-density III-nitride and III-V materials on lattice-mismatched substrates; this is also interesting from a fundamental viewpoint. Here, we develop a growth model for vertical nanowires which, under rather general assumptions on the solid-like coalescence process within the Kolmogorov crystallization theory, results in a morphological diagram for the asymptotic coverage of a substrate surface. The coverage is presented as a function of two variables: the material collection efficiency on the top nanowire facet a and the normalized surface diffusion flux of adatoms from the NW sidewalls b. The full coalescence of nanowires is possible only when a=1, regardless of b. At a>1, which often holds for vapor–liquid–solid growth with a catalyst droplet, nanowires can only partly merge but never coalesce into continuous film. In vapor phase epitaxy techniques, the NWs can partly merge but never fully coalesce, while in the directional molecular beam epitaxy the NWs can fully coalesce for small enough contact angles of their droplets corresponding to a=1. The growth kinetics of nanowires and evolution of the coverage in the pre-coalescence stage is also considered. These results can be used for predicting and controlling the degree of surface coverage by nanowires and three-dimensional islands by tuning the surface density, droplet size, adatoms diffusivity, and geometry of the initial structures in the vapor–liquid–solid, selective area, or self-induced growth by different epitaxy techniques.

## 1. Introduction

Semiconductor nanowires (NWs) offer almost unlimited possibilities for the bottom–up bandgap engineering and design of Si-integrated III-V optoelectronic device structures [1]. Vertical NWs can be grown by the vapor–liquid–solid (VLS) method [2,3,4] with a foreign catalyst metal, often Au [2], which can be replaced with a group III metal in the self-catalyzed VLS approach for III-V NWs [3]; selective area epitaxy (SAE) without any droplets [5]; or using the self-induced (SI) nucleation mechanism typical for III-nitride NWs on Si(111) substrates with different interlayers [6,7]. High aspect ratio NWs and other elongated structures are able to relax elastic stress induced by lattice mismatch on their strain-free sidewalls [8,9,10]. Therefore, they can be grown on dissimilar substrates such as Si without forming misfit dislocations [9]. If vertical NWs grow in both vertical and radial directions, they may merge together or even fully coalesce into continuous film. The coalescence process was studied in much detail for high density III-nitride NWs [11,12,13,14,15,16] and showed to yield nearly dislocation-free planar layers of GaN and AlGaN on Si [13,15,16]. The NW coalescence was also investigated for vertical InAs [17], ZnO [18], and Ge [19] NWs, where it can be desirable (for example, for obtaining continuous III-V films on Si with a possibly reduced dislocation density compared to planar epi-layers) or undesirable (for applications requiring isolated NWs). The coalescence process in dense ensembles of NWs and other elongated structures which are able to change their aspect ratio in different stages of epitaxial growth is also interesting from a fundamental viewpoint.

Here, we develop on the existing models for the vertical NW coalescence [11,20,21] by considering an ensemble of vertical NWs in the coalescence stage and describing the large time asymptote of the surface coverage as a function of the NW geometry (surface density, droplets on the NW tops, or flat tops without any droplets) and the diffusion length of adatoms on the NW sidewalls. We also consider the NW growth kinetics and the evolution of the surface coverage in the pre-coalescence stage. Our approach can also be used for studying the coalescence of three-dimensional (3D) islands or planar NWs in SAE [22,23], where the full coalescence of the initially separated structures is always required to obtain 100% coverage of a growth template [24,25,26]. Such a treatment was not performed so far to our knowledge and should be useful for understanding and controlling the late growth stages and coalescence process of different 3D structures in a wide range of material systems.

## 2. Model

We consider an ensemble of vertical NWs (or 3D islands) which grow in both vertical and horizontal directions with respect to the substrate surface. They nucleate with random positions and a surface density $N=P^{-2}$, where P is the average distance between the NW centers. For VLS or SAE growths, the NW density is determined by the pitch separating the pre-existing droplets or patterned pinholes in a dielectric mask, while in the SI growth of GaN NWs it depends on the nucleation mechanism of 3D islands which subsequently become NWs [7,11]. In the SAE process, the randomness in the NW positions is less than on non-patterned substrates. All NWs before coalescence have the same shape such that their cross-sectional area is given by $cR^{2}$, with R as the NW radius and c as a shape constant (c=π for cylindrical and $c=3\sqrt{3}/2$ for hexahedral NWs). Different stages of NW growth are illustrated in Figure 1. For VLS NWs with catalyst droplets on top, we consider the droplet contact angles β≤ 90°, in which case the two-dimensional (2D) projection of the droplet in the substrate plane is the same as for NWs. At β> 90°, droplets will start to merge earlier than NWs, and it is very difficult to say what happens to the VLS growth of NWs after that. Small contact angles of Ga droplets in the range from 50° to 90° are typical, for example, for thick Ga-catalyzed GaAs NWs with fast radial growth [3], and it is reasonable to assume that faster radial growth leads to smaller droplet contact angles in the general case.

The NW coalescence process should usually occur in a late stage of growth where the NW ensemble consumes the entire flux v of semiconductor material (group III atoms for III-V NWs) sent from vapor [27,28,29]. In molecular beam epitaxy (MBE), this corresponds to the full shadowing of the substrate surface which either re-emits material or contribute into the total diffusion flux coming to the NW tops [28]. In vapor phase epitaxy (VPE) techniques, this corresponds to the saturation of the total flux per NW which includes the growth species desorbed from different surfaces [29]. Introducing the average NW length L, we can write
(1)$$\frac{d}{dt}\left(\theta L\right)=v,$$
where θ is the surface coverage. In stage (a) of isolated growth where $\theta =cR^{2}N\ll 1$, with R as the average NW radius, this is equivalent to $d(cR^{2}N)/dt=v$, as in Refs. [28,29]. The growth regime described by Equation (1) starts at a certain NW length $L_{*}$ and coverage $\theta_{*}$ (most probably before coalescence, corresponding to $\theta_{*}=cR_{*}^{2}N$, with $R_{*}$ as the NW radius at the average length $L_{*}$ [28]). The pre-coalescence stage of NW growth will be discussed in the next section. Integrating Equation (1), we find
(2)$$\theta L=h,\;h=\theta_{*}L_{*}+v(t-t_{*}),$$
where $t_{*}$ is the moment of time after which the NW growth kinetics is governed by Equation (1). This equation shows that the volume of the NW ensemble per unit surface area θL scales linearly with time, and is proportional to the effective deposition thickness h.

For the vertical NW growth rate, we use the standard model of NW growth assuming that tall enough NWs elongate due to direct impingement of material onto their tops and surface diffusion of adatoms (group III adatoms for III-V or III-nitride NWs) from the upper part of NWs to their tops [4,28,30,31,32], re-written in terms of θ:(3)$$\theta \frac{dL}{dt}=v(a\theta +\Lambda u).$$

Here, a is the collection efficiency on the NW top, with a>1 for most VLS NWs having droplets on their tops due to a larger surface area of the droplet compared to $cR^{2}\;$ [4,33] or a higher pyrolysis efficiency at the liquid surface compared to solid [2], and a=1 for catalyst-free SAE or SI NWs. The dependence of a on the droplet contact angle in different epitaxy techniques will be considered in detail in the next section. In the late growth stage where the NW ensemble receives the whole vapor flux, we assume that a=const (corresponding to a fixed droplet contact angle), as in Refs. [28,29], while a is usually time-dependent in the preceding stages of growth. $\Lambda =\Lambda_{inc}$ denotes the effective collection length of adatoms on the NW sidewalls, limited by surface incorporation to allow for the radial NW growth, and u is the total NW perimeter per unit surface area. At$\;\theta =cR^{2}N\ll 1$ and u=2cRN, this is reduced to the usual growth law dL/dt=v(a+2Λ/R)  for isolated NWs [4,28,30,31,32]. NWs can grow infinitely long only when Λ≥0, which is the interesting case considered in what follows. At Λ<0, corresponding to “negative” diffusion of material from the NW tops to their sidewalls, NWs can only grow to a finite length [34] and should then fully coalesce due to the radial growth. The saturation of length at large t for Λ<0 is seen directly from Equation (3). The growth law given by Equation (3) is equivalent to
(4)$$\frac{dL}{dh}=a+\Lambda \frac{u}{\theta },$$
where u/θ is the average perimeter per unit cross-sectional area of NWs. This elongation law is valid at any θ, starting from θ≪1 to unity. In particular, the NW perimeter u tends to zero at θ→1. Therefore, we have L=ah at h→∞, which is consistent with Equation (2) at  θ→1 only when a=1. This gives a very simple answer to the main question in the title of this work. Vertical NWs can fully coalesce into continuous film only without any enhancement of the material collection at the NW top with respect to the NW sidewalls. In particular, VLS NWs with a>1 can partly merge, but will never coalesce into continuous film due to a magnifying effect of a catalyst droplet. This conclusion was drawn earlier in Ref. [28]; however, it was based on modeling the growth kinetics of isolated NWs. Any catalyst-free NWs or VLS NWs with a=1 will finally coalesce into continuous film regardless of the value of Λ, simply because surface diffusion becomes ineffective at u→0. For example, catalyst-free III-N NWs can fully coalesce [16].

To describe the degree of incomplete coalescence of NWs using Equations (2) and (4), we require a model for the NW perimeter per unit surface area u as a function of θ. According to the Kolmogorov crystallization model, the coverage θ is given by $\theta =1-exp(-\theta_{0})$, where $\theta_{0}$ is the so-called extended volume, or the coverage in the absence of coalescence [35]. The Kolmogorov model accounts for Poissonian nucleation with random position on a substrate surface, solid-like coalescence process which usually occurs for solid-state structures, and with neglect of the boundary effect. Applying this to the 2D projection of the NW ensemble in the substrate plane, with $\theta_{0}=cR^{2}N$, we have
(5)$$\theta =1-e^{-cR^{2}N},\;u=\frac{d\theta }{dR}=2cRNe^{-cR^{2}N}.$$

These expressions apply when the nucleation step is short compared to the duration of the whole NW growth process, which is guaranteed for most VLS [4] and SAE [5,36] NWs and is a good approximation for SI GaN NWs [7,11]. It is interesting to note that, due to Equation (2) for the total NW volume, we do not need to introduce any model for the NW radial growth rate, whereas the function $V\left(t\right)=dR/dt$ is essential in the Kolmogorov model [35]. Using Equation (5), we obtain
(6)$$u\left(\theta \right)=2\sqrt{cN}(1-\theta )\sqrt{-ln(1-\theta )}.$$

At small $\theta =cR^{2}N\ll 1$, this is reduced to $u=2\sqrt{cN\theta }=2cNR$ and $u/\theta =2\sqrt{cN/\theta }=2/R$. Figure 2 shows the normalized perimeters u and u/θ (in the units of $2\sqrt{cN}$) given by Equation (6), compared to the approximations $u/\left(2\sqrt{cN}\right)=\sqrt{\theta }$ and [$u/\theta ]/(2\sqrt{cN})=1/\sqrt{\theta }$ for isolated NWs at θ≪1. The curves for isolated NWs become inaccurate after θ≅0.1. The perimeter of NWs per unit area of the substrate surface u reaches its maximum at $\theta_{m}=1-1/\sqrt{e}≅0.39$, corresponding to the beginning of the coalescence process. The perimeter of NWs per unit area of their top facets  u/θ decreases monotonically with θ. This decrease is sharper than the simple $1/\sqrt{\theta }\;$ dependence for isolated NWs.

Using Equation (6) in Equation (4), we obtain the elongation law which accounts for the NW coalescence process:(7)$$\frac{dL}{dh}=a+b\left(\frac{L}{h}-1\right)\sqrt{-ln\left(1-\frac{L}{h}\right)},\;b=2\Lambda \sqrt{cN},$$
where h/L=θ according to Equation (2). Hence, the right-hand side can equivalently be presented as a function of θ. The numerical solution of this equation gives the mean NW length L, and simultaneously the surface coverage θ, as a function of h or the growth time. The generalized growth model given by Equation (7), together with a model for the surface coverage θ, is the main result of this work. In the next section, we analyze the asymptotic behavior of the coverage at h→∞ and L/h→const depending on the two control parameters, a≥1 and b>0, and consider the NW growth kinetics before the asymptotic stage.

## 3. Results and Discussion

First, we note that at $\theta =h/L=cR^{2}N\ll 1$, corresponing to low coverage and high NW aspect ratio, Equations (2) and (7) are reduced to the result of Ref. [28]:(8)$$\frac{dL}{dh}≅a+b\sqrt{\frac{L}{h}},\;R≅\sqrt{\frac{h}{cNL}}.$$

This special type of the Chini equation for L(h) can be resolved in the analytic form at a=const [28]. At h→∞, the NW length scales linearly with h: $L\to {\left(\sqrt{b^{2}+4a}+b\right)}^{2}/4$. Without coalescence, the coverage is given by $\theta =cR^{2}N$, and equals h/L according to Equation (8) for R. Therefore, the asymptotic coverage at h→∞ is given by [28]
(9)$$\theta_{\infty }≅\frac{4}{{\left(b+\sqrt{b^{2}+4a}\right)}^{2}}.$$

This result, obtained within a simplified model without coalescence, gives the asymptotic coverage correctly: $\theta_{\infty }=1\;$ only at a=1 and b=0.

The asymptotic coverage in the general case with coalescence is obtained from Equation (7) at L/h→const. Using L=ch in Equation (7) and finding the unknown c, the result is given by
(10)$$a\theta_{\infty }=1-b(1-\theta_{\infty })\sqrt{-ln\left(1-\theta_{\infty }\right)}.$$

This transcendent equation for $\theta_{\infty }$ cannot be analytically resolved but it enables one to plot the asymptotic coverage as a function of a and b. Figure 3 shows $\theta_{\infty }$ versus a for different b. As discussed above, the full coalescence of NWs occurs only at a=1 and regardless of b. For any a>1, NWs can only partly merge but will never fully coalesce. The asymptotic coverage monotonically decreases with a for any b, showing that a more efficient material collection at the NW top (enhanced by the presence of a catalyst droplet for VLS NWs) reduces the asymptotic surface coverage. For a given a>1, $\theta_{\infty }$ decreases with b. According to Equation (7) for b, the coverage is decreased for higher diffusivities of adatoms on the NW sidewalls having larger diffusion lengths Λ, and for larger surface densities N, or smaller pitches P in regular NW arrays. Decrease in the surface coverage and the corresponding increase in the NW aspect ratio for denser NW ensembles may look counter-intuitive at the first glance. However, this effect is well known, for example, for InAs NWs grown by MBE on patterned SiO_x_/Si(111) substrates [36], and is explained by the enhanced shadowing of denser NWs and lower material flux per NW contributing to the radial growth [28,29,37]. It does not contradict the fact that merging of NWs occurs earlier in denser NW ensembles, simply because the late stage of growth corresponding to the full shadowing of the substrate surface starts earlier than in sparse ensembles (corresponding to shorter times $t_{*}$ in Equation (1)). This effect will be considered in detail in what follows.

The approximation for isolated NWs given by Equation (9) predicts correctly some qualitative trends, in particular, the possibility of the full coalescence at a=1 (but only at b=0, which is wrong), and a decrease in the asymptotic coverage with b. However, it underestimates the coverage for any a at any positive b, that is, in the growth regimes with surface diffusion of adatoms. These inaccuracies, seen in Figure 3, are related to the overestimated perimeter per NW cross-sectional surface, as shown in Figure 2. In the growth model for isolated NWs, their total perimeter never decreases and never shrinks to zero. This gives a non-vanishing contribution of surface diffusion into the NW elongation rate, resulting in a higher NW aspect ratio and lower coverage of a substrate surface. In the refined model with the coalescence process included, the total NW perimeter reaches a maximum and then decreases to zero, leading to an almost negligible surface diffusion at large θ and zero diffusion flux of adatoms at θ→1.

Let us now consider the growth kinetics of a NW ensemble starting from the very beginning of growth. Re-formulating the results of Refs. [28,29] in terms of the coverage θ rather than $cR^{2}N$, which gives the coverage only at $cR^{2}N\ll 1$, we have
(11)$$\begin{array}{c} \frac{d\left(\theta L\right)}{dH}=\left(a\theta +Lu\right)\left(2-a\theta -Lu\right),\;a\theta +Lu\leq 1,\;\\ \frac{d\left(\theta L\right)}{dH}=1,\;a\theta +Lu>1, \end{array}$$
(12)$$\begin{array}{c} \frac{dL}{dH}=\left(a+\Lambda \frac{u}{\theta }\right)\left(2-a\theta -Lu\right),\;a\theta +Lu\leq 1,\;\\ \frac{dL}{dH}=a+\Lambda \frac{u}{\theta },\;a\theta +Lu>1. \end{array}$$

Here, H=vt is the 2D equivalent deposition thickness, and the parameters a and Λ have the same meaning as before. According to Equation (11), the volume of NWs per unit surface area increases due to the direct impingement of material onto the droplet surface (for VLS NWs with a>1 or a=1 depending on the droplet contact angle and the beam direction in MBE as discussed below) or flat tops (for catalyst-free SAE or SI NWs with a=1), and material collection on the sidewall surfaces of NWs having the total surface area Lu. The multiplying factor 2−aθ−Lu describes the contribution from the re-emitted species. This model applies when desorption of material (group III adatoms for III-V or III-nitride NWs) from the NW sidewalls is negligible, in which case all adatoms either diffuse to the NW top and contribute into the axial growth or get incorporated into the radial shell growing around each NW. Such growth occurs until a NW ensemble starts to collect the entire flux of the arriving material. After this moment of time, corresponding to aθ+Lu=1, the growth kinetics is governed by Equation (1). Equation (12), which actually apply for L≥Λ, with Λ as the incorporation-limited diffusion length of adatoms on the NW sidewalls, show that the axial NW growth rate decreases with its length due to a lower fraction of re-emitted species landing on each NW and becomes a+Λu/θ after saturation, as given by Equation (4).

These growth equations are written in the case of VPE with 100% pyrolysis efficiency at all surfaces [29], where
(13)$$a=\frac{2}{1+cos\beta },$$
with β as the contact angle of a catalyst droplet resting on the NW top (β=0 for non-VLS NWs). It is seen that a=2 at β= 90°, so that the NW top receives twice more flux than catalyst-free NWs without any droplet. Furthermore, a>1 for any β>0, showing that the magnifying effect of the droplet is always present in VPE-grown NWs. If the pyrolysis efficiency $\chi_{s}$ at the NW sidewalls is less than 100%, Equation (13) modifies to $a=\left(1/\chi_{s}\right)[2/\left(1+cos\beta \right)]$, so the a values larger than 2 are entirely possible even at β≤ 90°. In the directional MBE technique, the growth equations are modified [28]. In particular, the parameter a depends on the droplet contact angle β and the group III beam angle with respect to the substrate normal α according to [33]:(14)$$a=\frac{S(\alpha ,\beta )}{\pi R^{2}cos\alpha }.$$

Here, S(α,β) is the area normal to the beam intercepted by the droplet, which is given in Ref. [33]. The denominator $\pi R^{2}cos\alpha$ corresponds to 2D growth on the flat top facet. According to Ref. [33], S(α,β) is greater than $\pi R^{2}cos\alpha$ for $\beta >90^{{}^{\circ}}-\alpha$, corresponding to a>1. In particular, a=(1+1/cosα)/2 at β= 90°, which corresponds to the small stable contact angle of untampered Ga-catalyzed GaAs NWs for energetic reasons [38,39]. In this case, the magnifying effect of the droplet surface is present for all α>0, that is, for all beam directions except the one which is perpendicular to the substrate surface. The magnifying effect of the droplet disappears for small $\beta \leq 90^{{}^{\circ}}-\alpha$, where $\;S\left(\alpha ,\beta \right)=\pi R^{2}cos\alpha$ and hence a=1.

For both VPE and MBE NW growths, Equations (11) and (12) yield the evolution of the coverage with the average NW length in the following form:(15)$$\frac{d\theta }{dL}=u\frac{1-\Lambda /L}{a+\Lambda u/\theta },\;a\theta +Lu\leq 1.$$

We start the analysis with the simplest model for NW growth with negligible surface diffusion of adatoms and at a fixed contact angle of the droplet, corresponding to Λ=b=0 and a=const from the very beginning of growth. This model gives the lowest axial NW growth rate and hence the highest coverage compared to the general case with surface diffusion at b>0. Using $d\theta /dL=(d\theta /dR)\left(dR/dL\right)=udR/dL$, Equation (15) yields dR/dL=1/a for the NW radius. Therefore, R scales linearly with the NW length:(16)$$R=R_{0}+\frac{L}{a},$$
where $R_{0}$ is the initial NW radius at L=0. Using this in Equation (5), the surface coverage is obtained in the following form:(17)$$\theta =1-exp\left[-c{\left(R_{0}+L/a\right)}^{2}N\right],\;a\theta +Lu\leq 1.$$

The increase in coverage is described by this equation until the beginning of the late growth stage, which is governed by Equations (2) and (4). At Λ=b=0, these equations give the linear asymptotes $\theta L=\theta_{*}L_{*}+H-H_{*}$ and $L=L_{*}+a(H-H_{*})$, where $H_{*}$ is the deposition thickness at which the total flux per NW saturates at its maximum level. Therefore, further evolution of the surface coverage with the deposition thickness or NW length is given by
(18)$$\theta =\frac{\theta_{*}L_{*}+H-H_{*}}{L_{*}+a(H-H_{*})}=\frac{\theta_{*}L_{*}+\left(L-L_{*}\right)/a}{L},\;a\theta +Lu>1,$$
giving the asymptotic coverage $\theta_{\infty }=1/a$ at H→∞ or L→∞.

Figure 4 shows the evolution of the coverage with the NW length, obtained from Equations (17) and (18) at a fixed initial NW radius of 30 nm and a fixed inter-NW distance of 500 nm, for cylindrical NWs with c=π. The asymptotic coverage$\;\theta_{\infty }=1/a$ at Λ=0 depends only on a, regardless of the NW density, initial radius and the details of growth kinetics. From Figure 4, only NWs with flat tops (a=1) will fully coalesce. VLS NWs with a=1.5 and a=2 will only partly merge, but never coalesce into continuous film. The exponential increase in coverage given by Equation (16) to a much slower increase given by Equation (17) occurs at short NW lengths (from 190 to 210 nm in this example). Therefore, the slow increase in coverage in the late stage of NW growth provides a good approximation for nearly all growth times excluding the initial stage before the saturation of the total flux received by a NW ensemble. This observation also holds in the general case with surface diffusion of adatoms (b>0), because the diffusion-induced terms in the growth equations become negligible compared to the direct impingement at small enough u.

Figure 5 shows the asymptotic coverage of the substrate surface as a function of the droplet contact angle β for VLS NWs grown by different epitaxy techniques. The dependence a(β) for VPE growth method is obtained from Equation (13). For the directional MBE growth, it is calculated from Equation (14) using the expression of Ref. [33] for S(α,β) at two different beam angles of 45° and 30°. All these NWs will partly merge in the asymptotic growth stage. VPE-grown NWs will never fully coalesce, unless the droplets are removed from their tops. Conversely, MBE-grown NWs will fully coalesce for small enough droplet contact angles such that $\beta \leq 90^{{}^{\circ}}-\alpha$. Such VLS NWs are equivalent to catalyst-free NWs due to a=1 in both cases. As mentioned above, VLS growth without surface diffusion of adatoms yields the maximum possible coverage, because the axial NW growth rate increases and the radial NW growth rate decreases for larger diffusion lengths Λ. Therefore, the asymptotic coverage will be less than in Figure 5 for VLS NWs with surface diffusion.

Presenting θ as a function of R, Equation (15) with arbitrary Λ>0 takes the following form:(19)$$\frac{dR}{dL}=\frac{1-\Lambda /L}{a+\Lambda u/\theta }.$$

Using Equation (5) for θ and u and integrating, we obtain the analytic relationship between the NW radius (or the coverage θ) and length:(20)$$\frac{L}{\Lambda }-1-ln\left(\frac{L}{\Lambda }\right)=\frac{a\left(R-R_{0}\right)}{\Lambda }+ln\left(\frac{\theta }{\theta_{0}}\right).$$

Here, θ is given by Equation (5), $R_{0}$ is the NW radius at L=Λ, and $\theta_{0}=1-exp\left(-cR_{0}^{2}N\right)$ is the initial coverage at $R=R_{0}$. At Λ→0, this is reduced to the linear R(L) dependence given by Equation (16).

For high-temperature growths with enhanced desorption from the NW sidewalls, the NW top collects material from the upper NW section of length $\Lambda =\Lambda_{inc}$, while the desorption-limited diffusion length equals $\Lambda_{des}$ [29,37]. The radial NW growth can occur only when $\Lambda_{des}>\Lambda_{inc}$ [29]. In this case, Equation (15) rewrites [29]:(21)$$\frac{d\theta }{dL}=u\frac{\Lambda_{des}-\Lambda_{inc}}{L(a+\Lambda u/\theta )},\;a\theta +Lu\leq 1.$$

Presenting θ as a function of R and repeating the above procedure, the analytic relationship between the NW length and coverage modifies to
(22)$$\frac{L}{\Lambda_{des}}={\left(\frac{\theta }{\theta_{1}}\right)}^{\gamma }exp\left[\gamma a\frac{R-R_{1}}{\Lambda_{inc}}\right],\;\gamma =\frac{1}{\Lambda_{des}/\Lambda_{inc}-1}.$$

Here, $R_{1}$ is the NW radius at $L=\Lambda_{des}$, and $\theta_{1}=1-exp\left(-cR_{1}^{2}N\right)$ is the coverage at ${R=R}_{1}$.

SI GaN NWs on different substrates, emerging from the Volmer–Weber 3D islands, usually have very high surface density, up to 10^11^ cm^−2^ [6,7,11,32,40], which is why the coalescence process starts quite early. The typical base radius of the initial 3D islands equals 5 nm [7,32,37]. Desorption of Ga adatoms from the NW sidewalls should be present in high-temperature MBE growth of GaN NWs at around 800 °C [32]. According to Equation (22), in the initial stage of growth of separated GaN NWs, corresponding to $\gamma a\left(R-R_{1}\right)/\Lambda_{inc}\ll 1$ and $cR^{2}N\ll 1$, the NW length scales as a power law of its radius:(23)$$\frac{L}{\Lambda_{des}}≅{\left(\frac{R}{R_{1}}\right)}^{2\gamma },$$
with the power exponent γ=1.23 according to Ref. [7].

Figure 6 shows the coverage versus L for a model system with $R_{0}=$ 5 nm, $\Lambda_{des}=$ 40 nm [7,32], γ=1.23 [7], and $c=3\sqrt{3}/2$ for hexahedral NWs, at three different surface densities N= 10^10^, 5 × 10^10^ and 10^11^ cm^−2^, corresponding to the NW separations P= 100, 45 and 32 nm. The curves are obtained from Equation (22) at a=1 in the early stage and from numerical solution of Equations (2) and (4) after the flux saturation. These curves show the importance of the NW surface density (determined in the nucleation stage of 3D islands in the case of SI GaN NWs) in the kinetics of the coalescence process. The low surface density NWs (N= 10^10^ cm^−2^, P= 100 nm) remain separated even at a large L of 2000 nm, although they will finally coalesce due to a=1. The higher surface density NWs start to merge in the early stage of growth. The highest surface density NWs (N= 10^11^ cm^−2^, P= 32 nm) are almost fully coalesced at L= 2000 nm.

## 4. Conclusions

To summarize, the developed model predicts the following picture of the NW coalescence process. Any VPE-grown VLS NWs with the enhanced material collection efficiencies at their tops such that a>1 cannot fully coalesce even without any surface diffusion of adatoms from the NW sidewalls (at b=0), unless their droplets are removed (a=1) or the direction of the adatom diffusion flux is reversed (b<0). MBE-grown NWs can fully coalesce if the contact angle of their droplets meets the condition $\beta \leq 90^{{}^{\circ}}-\alpha$. At β≤ 90°, with the a values are typically less than 2. Therefore, merging or partial coalescence of VLS NWs is entirely possible according to Figure 3. Catalyst-free SAE or SI NWs with a=1 can coalesce into continuous film regardless of the strength of surface diffusion, because the adatom diffusion flux is disabled for large enough coverages θ. NWs with larger a and b will reach a lower asymptotic coverage. This corresponds to a lower coverage and higher aspect ratio of denser NWs and material systems with longer diffusion lengths of adatoms on their sidewalls. The coalescence process usually occurs in the late stage of growth where a NW ensemble collects all material sent from vapor. However, our model is capable of describing the NW growth kinetics from the early stage where NWs grow separately at low surface coverage or only start to merge. The model is purely geometrical and contains the characteristics of a concrete material system and epitaxy technique only in the control parameters a (the droplet contact angle for VLS NWs and the beam direction in MBE) and b (the pitch of a NW array and the diffusion length of adatoms on the NW sidewalls). Therefore, it can be applied to a wide range of material systems and growth technologies, including VLS and catalyst-free NWs of III-V, III-nitride and oxide materials and elemental semiconductors. The model is not even restricted to vertical NWs and can treat any 3D surface islands which are able to change their aspect ratio in the course of growth. We now plan to consider the time-dependent coalescence process in different ensembles of NWs (with a time-dependent $a\left(t\right)$ for VLS NWs) and other 3D structures and epitaxy techniques from the viewpoint of the obtained results. In particular, we plan to investigate the coalescence process in dense ensembles of SI GaN NWs grown by plasma-assisted MBE on Si substrates.

## Figures and Tables

**Figure 1 nanomaterials-13-02768-f001:**
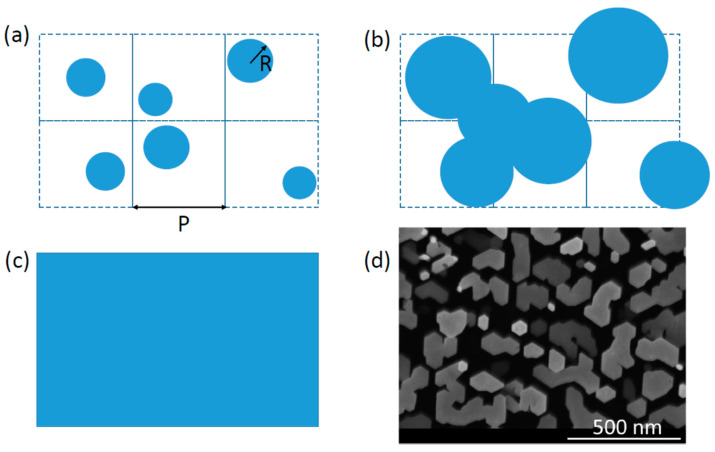
Stages of the growth process resulting in (**a**) isolated NWs, (**b**) partly merged NWs, and (**c**) fully coalesced NWs with θ=1. (**d**) Mosaic structure of merged SI GaN NWs [7] in stage (**b**). A similar picture of the coalescence process applies to 3D islands in SAE, where the islands emerge with a small size and a given density (determined in the nucleation step) but then merge together to fill a growth template.

**Figure 2 nanomaterials-13-02768-f002:**
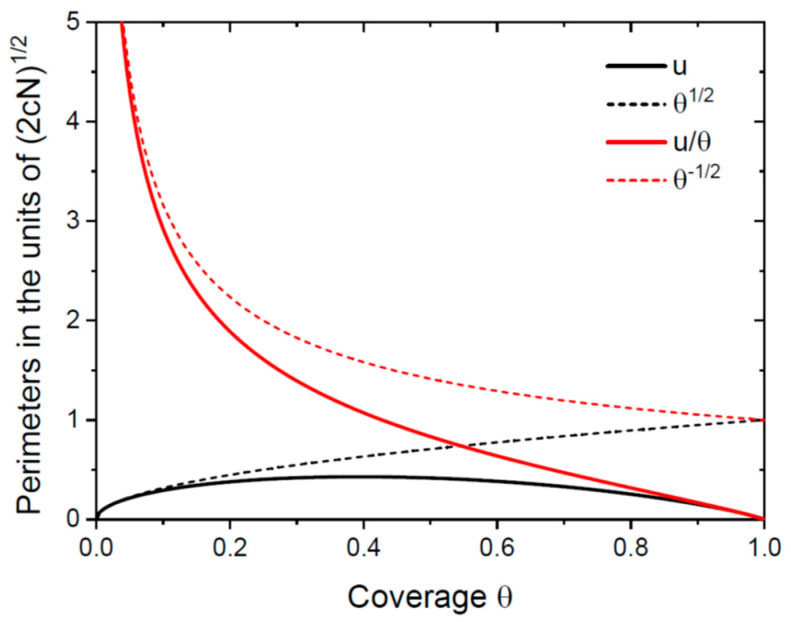
Normalized perimeters per unit surface area u and per surface area of the NW tops u/θ versus θ (the solid lines). The dashed lines show the same functions for isolated NWs.

**Figure 3 nanomaterials-13-02768-f003:**
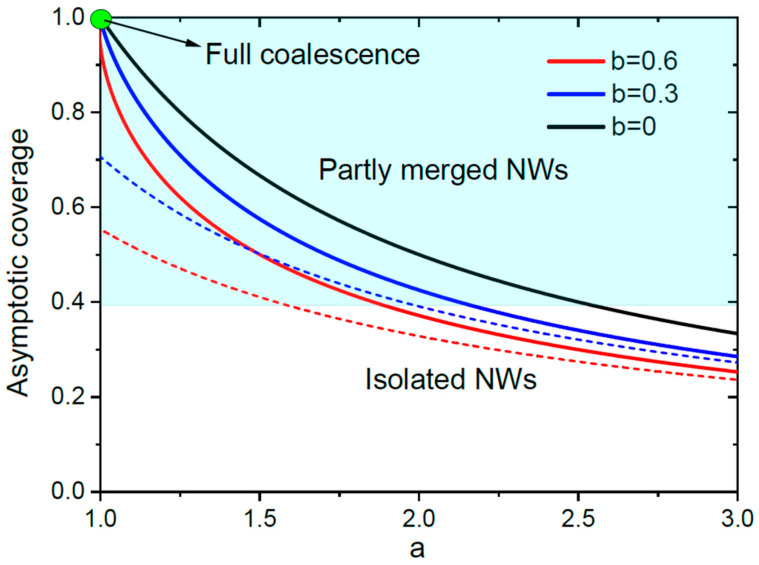
Asymptotic coverage $\theta_{\infty }$ versus a at three different b shown in the legend (the solid lines), obtained from Equation (10). The dashed lines show the approximations given by Equation (9) for isolated NWs, which underestimate the coverage for any a at b>0. At b=0, both models give the same result for $\theta_{\infty }=1/a$. The full coalescence occurs only at a=1. The asymptotic coverage at a fixed a>1 decreases with b. The shaded zone corresponds to partly merged NWs at $\theta_{\infty }>\theta_{m}=0.39$.

**Figure 4 nanomaterials-13-02768-f004:**
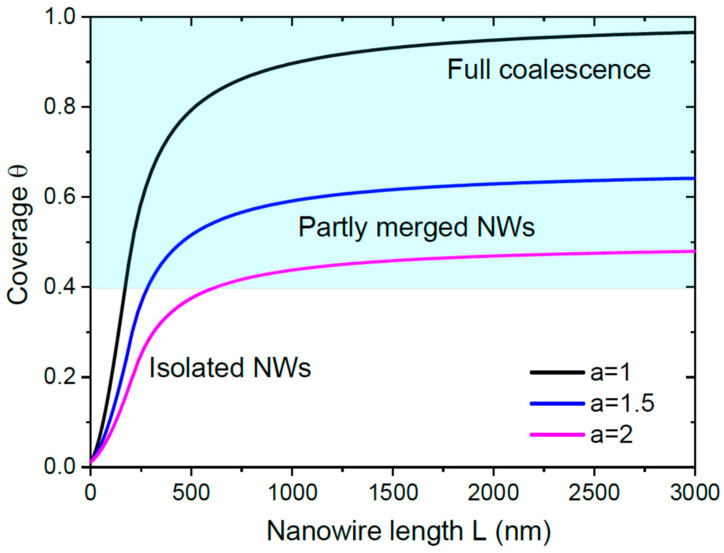
Maximum coverage of a substrate surface with NWs that start from a fixed radius of 30 nm and have a fixed surface density corresponding to an average separation of 500 nm, at different a shown in the legend. NWs will fully coalesce only at a=1, as in Figure 3. Other NWs will only partly merge.

**Figure 5 nanomaterials-13-02768-f005:**
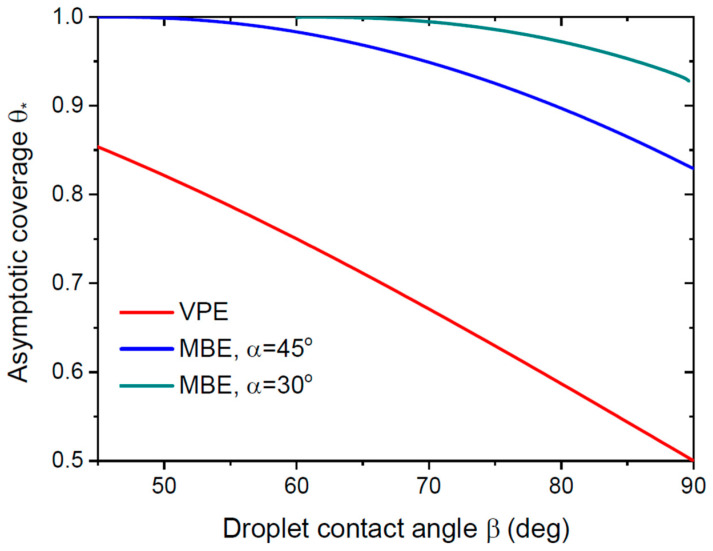
Maximum asymptotic coverage as a function of the droplet contact angle for VLS NWs grown by VPE and MBE at α= 45° and 30°. VPE-grown NWs will only partly merge but never coalesce into continuous film. MBE-grown NWs will fully coalesce at small enough droplet contact angles $\beta \leq 90^{{}^{\circ}}-\alpha$.

**Figure 6 nanomaterials-13-02768-f006:**
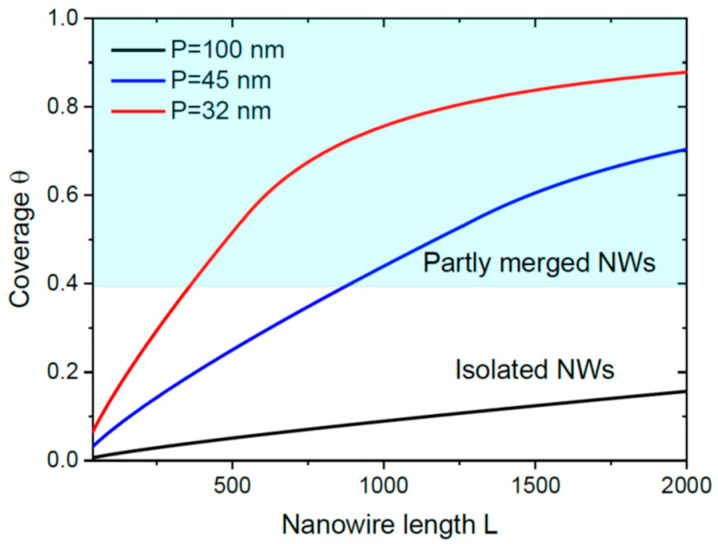
Evolution of the coverage with the length of catalyst-free NWs for three different NW surface densities corresponding to the average separations P shown in the legend. At a low surface density of 10^10^ cm^−2^ (P= 100 nm), NWs do not even start to merge at an average length of 2000 nm, while the higher surface density NWs with N= 5 × 10^10^ and 10^11^ cm^−2^ merge early. At the highest surface density of 10^11^ cm^−2^, the NW film almost reaches continuity at L= 2000 nm.

## Data Availability

Not applicable.

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
