# Peer review of "Can Nanowires Coalesce?"

_nanomaterials, 2023, doi:10.3390/nano13202768_

Round 1
Reviewer 1 Report
The author considers an interesting issue both from the theoretical and practical (as indicated already in the abstract and the introduction) perspectives, namely the possibility of coalescence of three-dimensional nanostructures, in particular of nanowires. The issue is certainly within the scope of MDPI Nanomaterials.
The main achievement of the paper is the formulation of a novel model for coalescence of semiconductor nanowires. The developed model may also be useful for studying other structures, thus it is of highly applicative nature.
The present contribution may be perceived as an important extension of previous achievements of the author, in particular of the papers [28] and [29], also published in MDPI Nanomaterials. Taking into account its potential usefulness e.g. in providing an in-depth outlook on the dynamics of wetting phenomena, this paper may be accepted "as it is".
Author Response
Response letter is attached.

Reviewer 2 Report
The paper is devoted to a theoretical description of the coalescence during the growth of nanowires (NWs). Two cases are distinguished depending on the value 'a' of the collection efficiency at the top of the NW. It is the ratio of the area from which the flux is collected to the top facet area of the NW. This ratio is equal to 1 for catalyst free growth and a>1 for the case of a droplet on the NW top where the droplet area is larger than the NW top area. The case a=1 results in the NW coalescing into a continuous film and does not raise any questions. It seems to me that the case a>1 is not treated properly. A constant value of a>1 during radial NW growth assumes radial growth of the droplet size at the same rate. Since there is no addition of droplet material during NW growth, the constant a>1 cannot be realised. Instead, the value of 'a' will decrease. As long as a>1, the droplet (in the horizontal section) will remain larger than the NW tip, and the radial NW growth will cause the droplets to coalesce before the NWs coalesce. The total area covered by the droplets will then increase (according to Kolmogorov's law) more slowly than the total area of the NWs. This process can lead to NW coalescence under a continuous film of droplet material. Alternatively, radial NW growth may result in the NW diameter being equal to the cross-sectional size of the droplet, reducing 'a' to 1 prior to NW coalescence. In any case, conclusions based on a constant value of a>1 during radial growth of NWs need to be reconsidered.
Author Response
Response letter is attached.

Round 2
Reviewer 2 Report
The author took into account my comments and improved the paper accordingly. The paper can be published in the present form.
In the equations containing fractions, e.g. eq. (4), the comma after the equations looks like a prime sign at the denominatior. Please add enough space in between.